# First Report of the Plasmid-mediated *fosB* Gene in *Enterococcus faecalis* from Pigs

**DOI:** 10.3390/genes12111684

**Published:** 2021-10-23

**Authors:** Xiaoming Wang, Yi Gao, Xiao Liu, Naiyan Sun, Jinhu Huang, Liping Wang

**Affiliations:** MOE Joint International Research Laboratory of Animal Health and Food Safety, College of Veterinary Medicine, Nanjing Agricultural University, Nanjing 210095, China; wangxm@njau.edu.cn (X.W.); dygaoyi@163.com (Y.G.); 15269293285@163.com (X.L.); 2019107023@njau.edu.cn (N.S.); jhuang@njau.edu.cn (J.H.)

**Keywords:** antimicrobial resistance, molecular genetic, fosfomycin resistant, *Enterococcus* spp.

## Abstract

Plasmid-mediated fosfomycin determinants is a global public health concern due to the increasing dissemination of fosfomycin resistance and limited clinical treatment options. Information about the fosfomycin resistant and molecular genetic among *Enterococcus* spp. is still lacking. In this study, we found the first plasmid-medieted *fosB* in *Enterococcus faecalis* from pigs, and all the fosfomycin resistant *Enterococcus* spp. (FRE) isolates were multi-drug resistant. S1-PFGE, Southern blot and conjugation experiments indicated that the *fosB* gene located on ~54.7 kb transferable plasmids_._ Relative competition assay confirmed that the *fosB**-*carrying plasmid impaired fitness in recipient *E. faecalis* JH2-2. Illumina and the MinION sequencing data revealed that both *E. faecalis* ES-1 and ES-2 isolates belonged to novel ST (ST964), and had 71 SNPs difference. WGS showed that the genetic environments of *fosB* were diverse among different species, and the linezolid resistance gene *optrA* was found in the *fosB*-carrying strains. To summarize, for the first time, we reported plasmid-mediated *fosB* in *E. faecalis* from pigs. And, the co-occurrence of *fosB* and *optrA* pose a serious threat to public health.

## 1. Introduction

*Enterococcus* spp. are natural inhabitants of the environment and essential components of the intestinal microbiota of healthy humans and animals [1,2]. The genus *Enterococcus* presently contains over 50 species, among which *Enterococcus*
*faecalis* (*E*. *faecalis*) and *Enterococcus*
*faecium* (*E**. faecium*) are the most predominant isolated species, accounting for more than 80% of isolates [3]. It has been recognized as important major nosocomial pathogens due to quickly acquiring virulence and multidrug resistance determinants via mobile elements [4]. However, the spread of fosfomycin resistant *Enterococcus* spp. (FRE), together with the limited availability of novel antimicrobial agents, has significantly restricted effective therapeutic options [5]. As a broad-spectrum cell wall synthesis inhibitor, fosfomycin interferes with the first step of bacterial cell wall biosynthesis, the formation of the peptidoglycan precursor UDP N-acetylmuramic acid, and then inhibits of enzyme-catalyzed reaction [6]. Therefore, it has considerable bactericidal activity against a range of bacteria by virtue of specific antimicrobial mechanisms [7].

Several mechanisms have been proposed to be related to fosfomycin resistance [8,9,10]. The mutations in the targeted enzyme MurA could reduce the affinity of enolpyruvyl transferase for fosfomycin [8], and mutations in chromosomal genes enconding fosfomycin transporters are also involved in causing resistance [9,10]. However, a significant challenge to the effectiveness of fosfomycin is the emergence of plasmid encoding enzymes that modify the antibiotic [11,12]. To date, four plasmid-encoded fosfomycin-enzymes (FosA, FosB, FosC and FosX) have been described [13]. Among the above enzymes, FosB, which catalyzes L-cysteine-fosfomycin, was currently the only known plasmid-brone fosfomycin-resistance determinant in *Enterococcus* spp. [14,15]. Reports on *fosB*-plasmid mediated fosfomycin resistance in *Enterococcus faecium* are available [16]. However, the knowledge of plasmid-brone *fosB* in *E. faecalis* is still unkown. Therefore, further research on the phenotype and genetic molecule of FRE and *fosB*-positive *E. faecalis* is urgently required. The purpose of this study is to investigate the antimicrobial resistance and molecular genetic of FRE from pigs.

## 2. Materials and Methods

### 2.1. Bacterial Isolation and Identification

365 health pig anal swab samples were collected in one commercial pig farms for one time in Jiangsu province, China, 2017. All the samples were screened on the mEI agar plate (Land bridge, Beijing, China) supplemented with 128 mg/L fosfomycin. The identification of bacterial species was performed using MALDI-TOF MS (BruKer Daltonik, Bremen, Germany), and then confirmed by 16S rRNA sequence analysis as described previously [17]. The presence of *fosA*, *fosB*, *fosC* and *fosX* encoding fosfomycin-enzymes were detected by PCR and followed by Sanger sequencing as described previously [10]. Primers used for the detection of fosfomycin resistance genes were listed in Appendix A. 

### 2.2. Antimicrobial Susceptibility Testing

The MICs of FRE to penicillin, ampicillin, vancomycin, erythromycin, tetracycline, marbofloxacin, florfenicol, rifampin, linezolid, valnemulin, streptomycin, gentamycin and fosfomycin were determined using the broth microdilution method according to the guidelines of the Clinical and Laboratory Standards Institute (document VET01-A4). The MIC results were interpreted according to CLSI and European Committee on Antimicrobial Susceptibility Testing (EUCAST) (http://www.eucast.org) (6 Decemeber 2020) guidelines.

### 2.3. S1-PFGE and Southern Blotting 

S1 nuclease-PFGE and Southern blotting were performed to locate *fosB* and determine plasmid size in *E. faecalis* as previously described [18]. In brief, strains were embedded in agarose gel plugs and digested by S1 nuclease (TaKaRa, Beijing, China). Specific *fosB* probe were used for gene localization by Southern blotting using the DIG High Prime DNA Labeling and Detection Starter Kit II (Roche Diagnostics). *Xba*I restricted DNA of *S. enterica* serovar Braenderup H9812 was used as DNA marker. This method allowed the detection and estimation of the size of large bacterial plasmids in the presence of genomic DNA using pulsed-filed gel electrophoresis (PFGE).

### 2.4. Conjugation Assay

The conjugation assay was described in detail in our previous study [19]. Briefly, the horizontal transferability of *fosB* was examined using wild-type strain as donor cell and *E. faecalis* JH2-2 as recipient cell. Transconjugants were confirmed by PCR targeting the *fosB* gene and cooperated with the results of MICs. The transfer frequency was calculated as the number of transconjugants per recipient. 

### 2.5. Fitness Experiment

The biological cost of the acquisition of *fosB*-carrying plasmid was investigated by in vitro competition assays [19]. Competition assay was used to assess the relative fitness of *fosB*-bearing transconjugant JH-ES-1 and JH-ES-2 against *E. faecalis* JH2-2 (the tester strain). The overnight cultures of competitor and the tester were mixed at a rate of 1:1 at 0 h and diluted in LB broth. Subsequently, the mixture was incubated for 24 h, and diluted in 1000-fold into LB broth at 37 ℃ for 24 h. The mixtures at both startpoint (0 h) and endpoint (24 h) were plated on LB plates without or with 128 mg/L fosfomycin and incubated at 37 ℃ for 12 h. The relative competitive fitness W was calculated using the formula W = (ln(Rf/Ri)) / (ln(Sf/Si)). Ri and Si indicate transconjugant and recipient cells at 0 h, while Rf and Sf indicate transconjugant and recipient cells at 24 h. Statistical analysis was carried out using GraphPad Prism 5.0.

### 2.6. Genome Sequence and Analysis

Genomic DNA of the isolates were extracted using the Magen DNA Purification Kit (Magen, Shanghai, China), following the manufacturer’s instructions. The samples were sequenced with two different platforms, comprising Illumina and the Oxford Nanoproe Technologies (ONT) MinION platform. Paired-end libraries were constructed and sequenced using the Illumina HiSeq 2500 System (Annoroad, Beijing, China). Sequencing reads were assembled de novo with the SPAdes 3.5 tool. A Rapid Barcoding Sequencing Kit was used to construct the libraries sequenced in a MinION device as previously reported. Guppy basecalling software (v2.2) was used to generate fast5 files harboring the 1D DNA sequence from fast5 files with only raw data in the tmp folder. High-quality complete plasmids were constructed by hybrid de novo assembly of Illumina short reads and nanopore long reads data using the Unicycler v0.3 tool. Reference sequences of antibiotic resistance genes were from database ARG-ANNOT [20]. The sequence type (ST) of *E. faecalis* was performed in silico by multilocus sequence typing (MLST) analysis using the published database [21].

### 2.7. Nucleotide Accession Numbers

The genome sequences of *E. faecalis* ES-1 determined in this study have been deposited in GenBank under the BioProject number of PRJNA609523. 

## 3. Results

### 3.1. Antimicrobial Susceptibility Profiles of Fosfomycin Resistant Enterococcus spp.

In total, 54 FRE isolates were obtained from 365 feces samples, of which 28 were *E. faecalis* isolates and the others were *E. faecium* isolates. The isolation rates of fosfomycin resistant *E. faecalis* and *E. faecium* from pigs were 7.67% and 7.12%, respectively, which were higher than 0.3% in *E. faecalis* and 4.9% in *E. faecium*, respectively, from a China teaching hospital [10]. Although epidemiology studies in this field is limited currently, our study suggested that, to some extent, fosfomycin resistant *Enterococcus* spp. in animals maybe more serious than that in human in China [10].

To investigate the antimicrobial susceptibility of FRE, MICs were determined for 11 classes of antimicrobial agents, including 13 kinds of antibiotics. As shown in Figure 1A,B, all those *E. faecalis* and *E. faecium* were resistant to erythromycin, tetracycline and fosfomycin, and all strains were susceptible to vancomycin and valnemulin (Appendix A). Additionally, a high frequency of resistance was observed against marbofloxacin (85.71% and 76.92%), florfenicol (96.42% and 88.46%), streptomycin (92.85% and 69.23%) and gentamycin (89.28% and 73.07%), followed by rifampin (46.42% and 42.30%) (Figure 1A,B). Moreover, fosfomycin resistant *E. faecalis* exhibited a low resistance rate to penicillin (3.57%) and ampicillin (3.57%), while fosfomycin resistant *E. faecium* showed a high resistance rate (penicillin, 42.30% and ampicillin, 76.92%). Inversely, fosfomycin resistant *E. faecalis* were highly resistant to linezolid (76.92%), but fosfomycin resistant *E. faecium* presented a low resistance rate (linezolid, 7.69%) (Figure 1A,B). According to the previous study, strains that are resistant to three or more classes of antimicrobial agents are considered as multiresistance [22]. Based on this criterion, all the FRE strains in this study can be determined as multiresistance (Figure 1C,D). More seriously, strains that were resistant to seven classes of antimicrobial agents took the largest proportions, and were up to 50% and 34.6% for *E. faecalis* and *E. faecium*, respectively. Although FRE exhibited multi-resistance, our results showed that they were all susceptible to vancomycin and valnemulin, which indicated that these two drugs may be used as specific medicines for infections caused by FRE strains. 

To explore the plasmid-mediated determinants in those strains for fosfomycin resistance, PCR was used to detect the plasmid encoding genes (*fosA*, *fosB*, *fosC* and *fosX*). Results showed that two *E. faecalis* isolates, named ES-1 and ES-2, were positive for *fosB*, being confirmed by Sanger sequence. Of note, it was found that most fosfomycin resistance *Enterococcus* isolates were negative for the known *fos* genes, indicated that novel fosfomycin resistance determinants or *murA* mutations may exist among these isolates [10], which requires further analysis.

### 3.2. Location and Transferability of fosB-Carrying Plasmid

S1-PFGE and southern blot confirmed that *fosB* was harboured on a ~54.7 kb plasmid in both ES-1 and ES-2, and there were two plasmids in each of those two isolates (Figure 2A,B). To evaluate the transferability of *fosB* in *E. faecalis* ES-1 and ES-2, conjugation assays was performed. Results showed that *fosB* gene successfully transferred from *E. faecalis* ES-1 and ES-2 to recipient strains *E. faecalis* JH2-2. The transfer frequencies of *fosB*-carrying plasmids were 1.21 ± 0.96 × 10^−5^ and 3.72 ± 3.24 × 10^−5^, respectively. Two transconjugants was obtained (named as JH-ES-1 and JH-ES-2), and presented resistance to fosfomycin, gentamicin and erythromycin (Table 1), which indicated that gentamicin and erythromycin resistance determinants co-transferred with *fosB*, being a potential hazard for public health.

### 3.3. Fitness Cost of the Transconjugant

*fosB*-carrying plasmid transferring into the recipient strain may impact the fitness of the host, thus the fitness cost of *fosB*-carrying plasmid in transconjugant was evaluated by competitive advantage. The in vitro competition assays showed that transconjugants JH-ES-1 and JH-ES-2 had a relative fitness value (W) of 0.78 and 0.73, respectively, when compared with its parental strain JH2-2 (Figure 3). The results further suggested that the acquiring of *fosB*-carrying plasmid imposed a fitness cost to transconjugant. 

### 3.4. WGS Analysis of fosB-Carrying Isolates

To investigate the molecular genetic characteristic of the two *fosB* positive strains, whole genome sequencing (WGS) analysis was performed. WGS analysis showed that the two isolates both belonged to novel ST type (ST964), and had 71 SNPs. This indicated that clonal dissemination with novel ST type emerged in *fosB*-carrying *E. faecalis* strains. Additionally, one complete chromosome and two plasmids sequences were obtained from each one strain, and *fosB* gene was located on a 53.121-kb plasmid in both strains, which were correspondence with the S1-PFGE results. The *fosB*-carrying plasmid pES-fosB had 57% query cover and 97.68% identity with *E. faecalis* plasmid pKUB3007-3 (accession no. AP018546.1). Both the two plasmids carried *fosB*, macrolide resistance gene *erm*(B) and toxin-antitoxin (TA) systems (Figure 2C). Additionally, the pES-fosB plasmid also contained cadmium resistance gene *cadA*, copper chaperone (CopZ) and IS members of IS*Efm*, IS*1216E* and ΔIS*1216* (Figure 2C).

Analysis of the genetic environment of *fosB*-carrying contigs from NCBI showed that *fosB* in *E. faecium* was flanked by *tnpA*. And plasmid pEMA120 (genbank: KX853854) possessed a Tn*1546*-like element, with *tnpA* and *fosB* genes inserted in the *vanRS*-*vanH* intergenic region (Figure 4). The locations of the insertion in this region were identical, corresponding to nucleotide 5813 bp of Tn*1546* (GenBank M97297). *fosB* gene also emerged in *Staphylococcus haemolyticus* and *Staphylococcus aureus* in small plasmids, and flanked by rep gene at right end (Figure 4). However, *fosB* was bracketed by *hp* in *E. faecalis*, and then surrounded by *erm*(B) and *aac(6’)-aph(2’’)* (Figure 4). Those analyses indicated that the genetic environments of *fosB* were diverse.

The second plasmid in ES-1 and ES-2 named pES-erm(B), which has a size of 85.096-kb, and carried *erm*(B), chloramphenicol resistance gene *cat*, tetracycline resistance genes *tet*(L) and *tet*(M), TA systems, IS*1216E* and ΔIS*1297*, copper ATPases (CopA and CopB), and copper transport repressor (CopY) (Appendix A).

Further analysis of WGS showed that one copy *optrA* gene, which conferred resistance to oxazolidinones and phenicols, was in chromosome. Macrolide resistance gene *erm*(A) and florfenicol resistance gene *fexA* were flanked on the left of *optrA*, and the 11.739 kb (*tnpB*-*erm*(A)-*fexA-optrA-hp*) segment has 100% query cover and 99.81% identity with *E. faecalis* 743142 (accession no. MF443377.1), which located on plasmid. Partial segment containing *fexA*-*optrA* in *E. faecalis* ES1 also emerged in homo *E. faecalis* plasmid from China and Ireland with accession no. MH018572.1 and MN831417.1, respectively, as well as homo *E. faecalis* chromosome (accession no. MH225421.1) (Appendix A). Additionally, chromosome sequence also contained aminoglycoside-modify enzyme gene *aph(3’)-III*, lincosamide and streptogramin A resistance gene *lsa*(A), sulphonamide resistance gene *dfrG* and macrolide-lincosamide-streptogram B resistance gene *erm*(B). The emergence of multidrug-resistant genes will render antibiotic treatment ineffective.

## 4. Discussion

In this study, fosfomycin resistance in clinical *Enterococcus*. spp from health pigs in Jiangsu Province, China, was evaluated. As previously reported, fosfomycin has good activity for the treatment of complicated *Enterococcus*. spp infection [15]. However, in this study, all the FRE strains were multi-resistance, and the fosfomycin resistance rate was higher than that in the previous report [15]. According to our results, strains resistance to seven classes antimicrobial agents accounted for the largest proportion. Those indicated that fosfomycin should not usually be suggested in the treatment of *Enterococcus*. spp infection at least in some commercial pig farms, and it is necessary to monitor the antimicrobial resistance rate in FRE. Additionally, our results showed that all the FRE in this study were susceptible to vancomycin and valnemulin, which means that these two drugs could inhibit FRE growth at a low concentration, therefore, they have the potential to be used as specific medicines for infections caused by FRE strains.

The complete sequences of two *fosB*-carrying *E. faecalis* was obtained by WGS. The two strains shared the same novel ST964 with 71 SNPs difference. It was noting that clonal dissemination emerged in *fosB*-carrying *E. faecalis* strains. Additionally, competitive assay showed that *fosB*-carrying plasmid brought a fitness cost to the recipient strain. This finding partially explained why *fosB* is unfrequently detected in *E. faecalis.* The comparison of pES-fosB and related *E. faecalis* strains (genebank: AP018546.1) revealed that their plasmid backbones were relatively conserved. The sequence surrounding the *fosB* gene was diverse among different species. For example, the *rep* genes were adjacent to *fosB* in staphylococcal plasmid, while a *tnpA* gene existed in the downstream of *fosB* and was included in an IS*L3-like* transposon in *E. faecium* plasmid, [23]. However, *hp* genes located up- and downstream of *fosB* in *E. faecalis* strains, and we speculated that recombination of the *fosB* gene may have occurred when it was transferred to *E. faecalis*. Previous study demonstrated that *fosB* and *vanRS* co-located on the same plasmid [22]. And in our study, *fosB*, *erm(B)* and *aac(6’)-aph(2’’)* emerged on one single plasmid, this may be the results of cross-use antimicrobial agents. Since the *fosB*-carrying plasmid pKUB3007-3 hasn’t been reported until now, this is the first report of transferable *fosB*-carrying plasmids in *E. faecalis*.

The presence of two copies of *erm(B)* on pES-erm(B) plasmid was detected, and one copy *erm*(B) located on pES-fosB plasmid. The complicated existence of *erm*(B) could explain why ES-1 and ES-2 had high MIC values of erythromycin. Additionally, the coexistence of *fosB* and *optrA* in one single *E. faecalis* strain was firstly found, which was consistent with the bacterial drug resistance phenotype. As we know, essential trace elements such as chromium (Cr), zinc (Zn), and copper (Cu) participate in a number of biological processes and are important constituents of several key enzymes being involved in several oxidation-reduction reactions [24,25]. However, it would become harmful if excessive feeding was applied. In this study, cadmium resistance gene *cadA*, copper transport related protein CopZ/B/Y were incorporated into plasmid pES-erm(B) and pES-fosB, which hinted that these genes may be spread to divergent bacteria by horizontal transfer [26]. Additionally, trace elements related genes emerged in transferable plasmid, which may be induced by feed additive in livestock diets. These findings indicated that excessive feeding could be harmful in animal production through inducing the spread of trace elements.

In conclusion, we firstly reported the emergence of plasmid-mediated fosfomycin resistant gene *fosB* in *E. faecalis* strains with novel ST964 from pigs, and also studied the antimicrobial resistance and molecular genetic of fosfomycin resistant *Enterococcus* spp.. The *fosB-*carrying plasmid could transfer between different strains, which lead to decreased sensitivity to multiple antibiotics in the recipient bacteria. *fosB*-carrying plasmid brought fitness cost in recipient *E. faecalis* JH2-2, which partially explained the low detection rate of *fosB* in *E. faecalis.* The genetic environments of *fosB* were diverse in different species, and *fosB* could coexist with *optrA*, which has a potential threat to public health. Given that fosfomycin and linezolid are the last-resort antibiotic for treating infections caused by VRE, the co-occurrence of *fosB* and *optrA* in one single *E. faecalis* may seriously compromise the effectiveness of clinical therapy. This finding, taking a “One Health” perspective, provided an importance theory to the increasing fosfomycin resistance. Continuous monitoring will be necessary to prevent further dissemination of these resistant elements in *Enterococcus* spp. isolates from humans and food producing animals.

## Figures and Tables

**Figure 1 genes-12-01684-f001:**
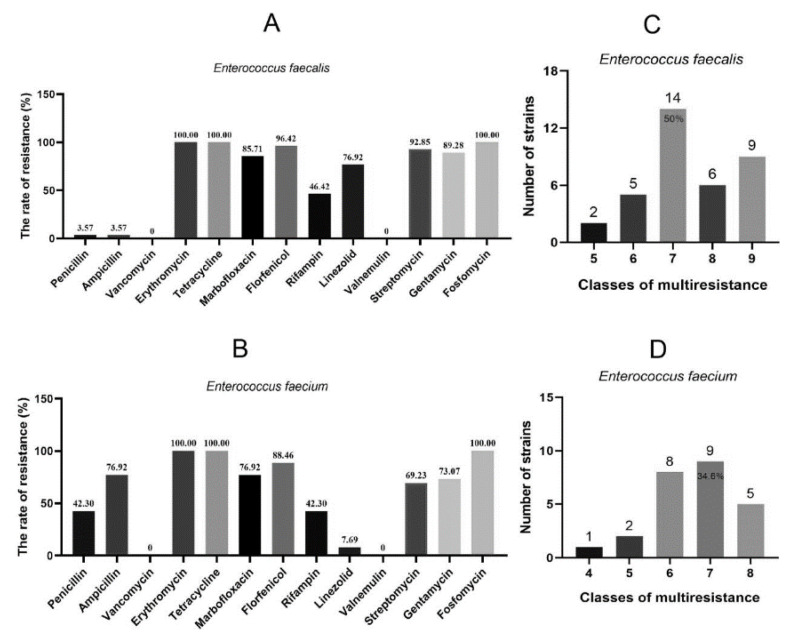
Antimicrobial resistant profiles of FRE strains. (**A**,**B**), the antibiotic-resistance rate of each tested drug; (**C**,**D**), the proportion of multiresistance FRE.

**Figure 2 genes-12-01684-f002:**
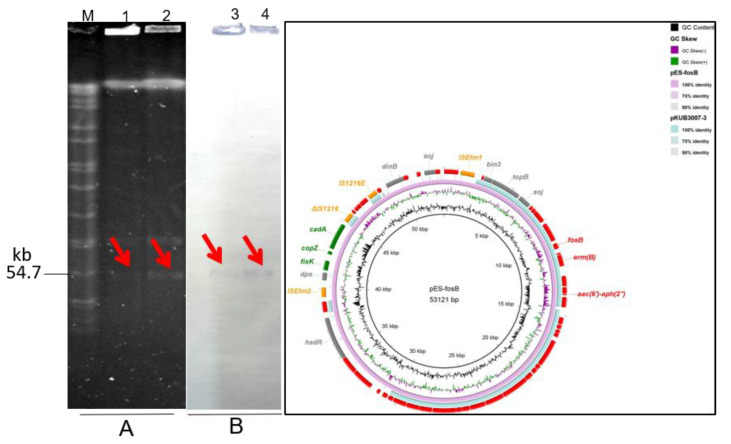
The location of *fosB* in wild-type *E. faecalis* strains ES-1 and ES-2. (**A**) S1-PFGE of *fosB*-carrying wild-type *E. faecalis* strains (**B**) the corresponding Southern hybridization using the *fosB*-specific probe. Lane M, marker H9812; lanes 1 and 2 represent S1-PFGE patterns of isolates ES-1 and ES-2, lanes 3 and 4 represent southern blot patterns of *fosB* gene in ES-1 and ES-2. (**C**) The genetic contents of plasmid pES-*fosB*. Circular representation of alignments between reference *fosB* carrying plasmid pKUB3007-3 (accession no. AP018546.1) and the homologous plasmid from *E. faecalis* ES1. The alignments were generated by Blastn and visualized by BLAST Ring Image Generator.

**Figure 3 genes-12-01684-f003:**
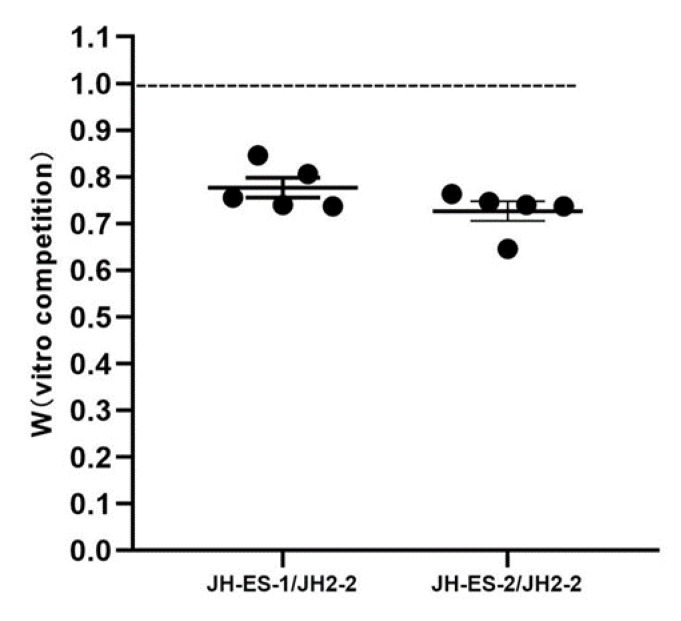
The relative competitive fitness W of the donor JH-ES-1 and JH-ES-2 with the recipient JH2-2 in vitro competition assays. The values were represented as mean ± SD of 5 independent parallels.

**Figure 4 genes-12-01684-f004:**
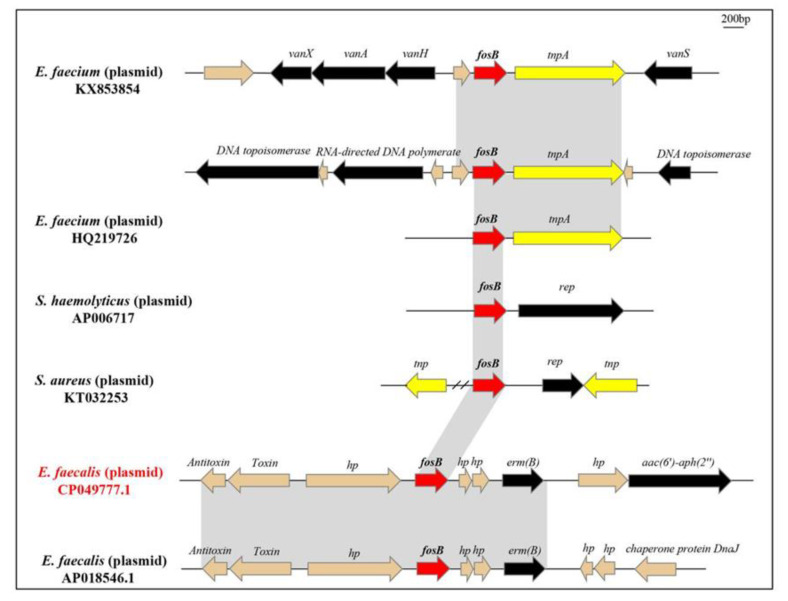
The *fosB* gene and genetic environment of *fosB* in different species or genera. *fosB* genes were indicated by red arrows, and regions of 100% nucleotide sequence identity were shaded in grey.

**Table 1 genes-12-01684-t001:** Characterization of *fosB* positive strains and their transconjugants.

Antimicrobial Agents (mg/L)/Other Features	ES-1	ES-2	JH2-2	JH-ES-1	JH-ES-2
Fosfomycin	2048	2048	64	512	512
Linezolid	4	8	0.5	0.5	0.5
Penicillin	4	2	1	1	1
Vancomycin	2	4	2	2	2
Gentamicin	HLGR	HLGR	-	HLGR	HLGR
Streptomycin	HLSR	HLSR	-	-	-
Florfenicol	64	64	2	2	2
Tetracycline	128	128	2	2	2
Erythromycin	>256	>256	0.5	256	256
Fusidic acid	4	4	>256	>256	>256
Rifampicin	2	2	>256	>256	>256
ST	964	964			
Resistance genes	*aph(3′)-III*, *dfrG*, *erm*(A), *erm*(B), *fexA*, *lsa*(A), *optrA*, *sul*, ***aac(6******′******)*****-*aph(2******′′******)*****, *cat*, *erm*(B), *fosB*, *tet*(L), *tet*(M)**	*aph(3′)-III*, *dfrG*, *erm*(A), *erm*(B), *fexA*, *lsa*(A), *optrA*, *sul*, ***aac(6******′******)*****-*aph(2******′′******)*****, *cat*, *erm*(B), *fosB*, *tet*(L), *tet*(M)**		** *aac(6* ** ** *’* ** ** *)* ** **-*aph(2*** ** *’’* ** ** *)* ** **, *erm*(B), *fosB***	** *aac(6* ** ** *’* ** ** *)* ** **-*aph(2*** ** *’’* ** ** *)* ** **, *erm*(B), *fosB***

Note: MICs shaded grey represented strains that were resistant to the corresponding antimicrobial agents; Antimicrobial resistance genes located on plasmids were shown in bold. HLGR, high-level gentamicin resistance; HLSR, high-level streptomycin resistance; –, not HLGR/HLSR. MICs of gentamicin and streptomycin > 500 ug/ml and 2000 ug/ml were defined as HLGR and HLSR, respectively, based on the CLSI.

## Data Availability

The data presented in this study are available on request from the corresponding author. The data are not publicly available due to privacy/ethical restrictions.

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
