# Peer review of "First Report of the Plasmid-mediated fosB Gene in Enterococcus faecalis from Pigs"

_genes, 2021, doi:10.3390/genes12111684_

Round 1

Reviewer 1 Report

Authors first proposed the plasmid-mediated fosB gene in Enterococcus faecalis from pigs. This report is interesting. However, some major issues should addressed.

Mojor

  1. Authors should provide a description of what further improvements were made through this study in abstract.
  2. The authors should add a description of the difference between Faecalis and Faecium strains. An explanation is required for why the strain was chosen for this study.
  3. The author needs to further explain in the discussion part that the results of this study mentioned in Result 1 can be used as specific drugs.
  4. In conclusion, the authors should add explanations about the implications of this study. Since this is the first report, it would be good to emphasize the importance of the study a little more.

Minor

  1. In Table 1, the author expressed gentamycin as HLGR, but it is necessary to explain the criteria for using this word when it is higher than that.
  2. It looks good if the plasmid is centered in Figure 2c.
  3. In Figure 2b, the resolution of the Southern blot results is low.
  4. Fixed typo in line 182 – study
  5. Line 298 requires alignment.
  6. In Figure 2, each lane should be marked in A and B.
  7. In line 154, the notation of ES1 and ES2 is different from the previous one.
  8. line 201 You need a rationale for your expectation from this content.

Author Response

Dear reviewer,

Thanks a lot for sending these favorable comments on our manuscript genes-1416258 (First report the plasmid-mediated fosB gene in Enterococcus faecalis from pigs). The helpful suggestions by you have all been incorporated into the revised version.

Major

  1. Authors should provide a description of what further improvements were made through this study in abstract.

Response: Thanks a lot for your constructive suggestions. We have further improved our description in abstract. Lines 8-20.

  1. The authors should add a description of the difference between Faecalis and Faecium strains. An explanation is required for why the strain was chosen for this study.

Response: The genus Enterococcus presently contains over 50 species, and E. faecalis and E. faecium are the predominant isolated species, accounting for more than 80% of isolates. E. faecalis was positive for the biochemical reaction of nitrite and pyruvate, but E. faecium was negative, however, it is contrary of arabinose. Additionally, on the blood plate, there was a significant difference between the two species bacteria. E. faecalis (cultured for 24 hours) formed small, pinhead sized colonies, raised and dry, while E. faecium formed medium and slightly smaller colonies. The description of these has added in Manuscript. Lines 24-28.

   Considering Enterococci serve as important key indicator bacteria for several human and veterinary resistance surveillance systems, we selected this specie strains to study.

  1. The author needs to further explain in the discussion part that the results of this study mentioned in Result 1 can be used as specific drugs.

Response: Because results of MIC indicated that all the FRE in this study were susceptible to vancomycin and valnemulin, and this showed that the two drugs could inhibit FRE growth at a low concentration, therefor, they may be used as specific medicines for infections caused by FRE strains. The further explain has added in the discussion part line 190-193.

  1. In conclusion, the authors should add explanations about the implications of this study. Since this is the first report, it would be good to emphasize the importance of the study a little more.

Response: Thank you for your constructive comments.We have add explanations about the implication of this study in conclusion. Line 226-240.

Minor

In Table 1, the author expressed gentamycin as HLGR, but it is necessary to explain the criteria for using this word when it is higher than that.

Response: MICs of gentamicin and streptomycin > 500 ug/ml and 2000 ug/ml were defined as HLGR and HLSR, respectively, based on the CLSI. Lines 106-107.

It looks good if the plasmid is centered in Figure 2c.

Response: Figure 2b is the southern blot of Figure 2a, and they should be arranged together.

In Figure 2b, the resolution of the Southern blot results is low.

Response: After trying southern blot for more than three times, the bands of fosB were indistinct, and we have readjust the resolution of the figure.

Fixed typo in line 182 – study

Response: It has been corrected accordingly. Line 185.

Line 298 requires alignment.

Response: It has been aligned accordingly.

In Figure 2, each lane should be marked in A and B.

Response: In Figure 2, each lane has been marked with M, 1-4 in A and B.

In line 154, the notation of ES1 and ES2 is different from the previous one.

Response: It has been renamed as ES-1 and ES-2. Line 162.

line 201 You need a rationale for your expectation from this content.

Response: In our study, hp genes located up- and downstream of fosB in E. faecalis strains, and we speculated that fosB gene may have occurred recombination when it was transferred to E. faecalis. Line 203-206.

Reviewer 2 Report

This study provides important information which could contribute to understand the role of plasmid-mediated fosB in the sensitivity of multiple antibiotics in E. faecalis. Overall, the manuscript is need comprehensive edit. some of the amendments are needed as following:

Line 2:  First report (of) the plasmid-mediated fosB gene in Enterococcus faecalis from pigs

Line 26 to 28: please add a reference.

Line 37: please add a reference.

Line 45: please add a reference

Line 142-144 and Line 157-163, This is belonging to the discussion not the result section

Line 180: from health pigs???

Line 183: multiresistance to multi-resistance

The references need some work – Please check the relevant section in the instructions for authors for more details

Figure 2: images are blurry, and the bands are not clear. What’s Lane M and which one lane 1-3. Please add more information.

English grammar requires improvement throughout the manuscript and many parts of the manuscript are poorly written such as Line 135 to 136, Line 33 to 35, Line 184-186 and Line 37-40.

Author Response

Dear reviewer,

Thanks a lot for sending these favorable comments on our manuscript genes-1416258 (First report of the plasmid-mediated fosB gene in Enterococcus faecalis from pigs). The helpful suggestions by you have all been incorporated into the revised version.

Comments and Suggestions for Authors

This study provides important information which could contribute to understand the role of plasmid-mediated fosB in the sensitivity of multiple antibiotics in E. faecalis. Overall, the manuscript is need comprehensive edit. some of the amendments are needed as following:

Response: Thanks a lot for sending these favorable comments on our manuscript genes-1416258 (First report the plasmid-mediated fosB gene in Enterococcus faecalis from pigs). The helpful suggestions by you have all been incorporated into the revised version.

Line 2:  First report (of) the plasmid-mediated fosB gene in Enterococcus faecalis from pigs

Response: It has been corrected accordingly. Line 2

Line 26 to 28: please add a reference.

Response: It has been added accordingly.

Line 37: please add a reference.

Response: It has been added accordingly.

Line 45: please add a reference

Response: It has been added accordingly.

Line 142-144 and Line 157-163, This is belonging to the discussion not the result section.

Response: These parts has moved to discussion. Lines 208-210, 215-219 and 222-225.

Line 180: from health pigs???

Response: yes, all the strains in our study from commercial health pigs.

Line 183: multiresistance to multi-resistance

Response: It has been added accordingly. Line 185.

The references need some work – Please check the relevant section in the instructions for authors for more details

Response: Ok, thank you.

Figure 2: images are blurry, and the bands are not clear. What’s Lane M and which one lane 1-3. Please add more information.

Response: After trying southern blot for many times, the bands of fosB were indistinct, and we have readjust the resolution of the Figure. In Figure 2, Lane M, marker H9812; lanes 1 and 2 represent S1-PFGE patterns of isolates ES-1 and ES-2, lanes 3 and 4 represent southern blot patterns of fosB gene in ES-1 and ES-2. Lines 118-120.

English grammar requires improvement throughout the manuscript and many parts of the manuscript are poorly written such as Line 135 to 136, Line 33 to 35, Line 184-186 and Line 37-40.

Response: The manuscript has been edited by native English.
